# Shoe-Integrated Sensor System for Diagnosis of the Concomitant Syndesmotic Injury in Chronic Lateral Ankle Instability: A Prospective Double-Blind Diagnostic Test

**DOI:** 10.3390/nano13091539

**Published:** 2023-05-04

**Authors:** Yanzhang Li, Rui Guo, Yuchen Wang, Jingzhong Ma, Xin Miao, Jie Yang, Zhu Zhang, Xiaoming Wu, Tianling Ren, Dong Jiang

**Affiliations:** 1Department of Sports Medicine, Peking University Third Hospital, Institute of Sports Medicine of Peking University, Beijing 100191, China; 2Beijing Key Laboratory of Sports Injuries, Beijing 100191, China; 3Engineering Research Center of Sports Trauma Treatment Technology and Devices, Ministry of Education, Beijing 100191, China; 4Beijing National Research Center for Information Science and Technology (BNRist), School of Integrated Circuits, Tsinghua University, Beijing 100084, China; 5Tsingyan Micro Technology Co., Ltd., Shaoxing 312099, China

**Keywords:** sensor system, diagnosis, graphene piezoelectric sensors, biosensing application, sports injury

## Abstract

Chronic lateral ankle instability (CLAI) is commonly secondary to prior lateral ankle ligament injury, and the concomitant latent syndesmosis injury would prolong recovery time and increase the risk of substantial traumatic arthritis. However, differentiating syndesmotic injury from isolated lateral ankle ligament injury in CLAI cases is difficult by conventional physical and radiological examinations. To improve the accuracy of syndesmotic injury diagnosis, a shoe-integrated sensor system (SISS) is proposed. This system measures plantar pressure during walking to detect the presence of syndesmotic injury. The study included 27 participants who had ankle sprains and underwent an examination. Plantar pressure in eight regions of interest was measured for both limbs, and syndesmotic injuries were examined using arthroscopy. The width of the syndesmosis was measured to evaluate its severity. The characteristics of plantar pressure were compared between patients with normal and injured syndesmosis. The results indicated that peak plantar pressure ratios with logistic regression predicted value > 0.51 accurately distinguished concomitant syndesmotic injury during walking, with high sensitivity (80%) and specificity (75%). The post-test probability of having a syndesmotic injury was positively 80% and negatively 25%. These findings demonstrate the effectiveness of cost-effective wearable sensors in objectively diagnosing concomitant syndesmotic injuries in cases of CLAI.

## 1. Introduction

Ankle sprains are among the most common injuries related to sports, accounting for nearly 40% of all such injuries, with lateral inversion sprains being the majority [1]. They affect people of all skill levels, from amateur to professional players. While many ankle sprains can heal with appropriate rest and rehabilitation, it is still possible for 20–40% of them to result in chronic lateral ankle instability (CLAI), causing lingering symptoms such as ankle pain, swelling, and instability [2,3]. Individuals with CLAI often experience the unsettling feeling of their ankle giving way, which can lead to further injuries such as fractures and dislocations. The concomitant syndesmosis injury is one of the conditions that sometimes be associated with CLAI [4]. This injury affects a group of ligaments that connect the tibia and fibula bones in the ankle joint, which are crucial for maintaining stability [5]. Injury to the syndesmosis is often overlooked in patients with CLAI, especially if the sprain is not severe, thus exacerbating instability, prolonging rehabilitation time, and increasing the risk of future traumatic arthritis [6].

Accurately diagnosing and distinguishing a syndesmotic injury from an isolated lateral ankle ligament injury is essential for the effective treatment of CLAI. However, conventional physical and radiological examinations may not always suffice in identifying the presence and severity of the syndesmotic injury [7]. Physical examinations are largely dependent on the expertise of medical practitioners and may be less effective for chronic ankle injuries [8]. Radiological data can exhibit ambiguous clinical correlation and are often overemphasized in clinical decision-making [9]. Arthroscopic exploration, the gold standard for diagnosis [10], is invasive and impractical for routine examinations. In this context, functional diagnostic tools, such as plantar pressure examination [11], have the potential to offer innovative approaches to diagnosis. For instance, Nyska [12] observed a significant reduction in relative forces under the heel and toes and a corresponding increase under the midfoot and lateral forefoot in subjects with chronic ankle instability. However, plantar pressure in CLAI with concomitant syndesmosis injury cases has not been studied. Traditional plantar pressure testing equipment has limitations, including a long detection time and difficulty interpreting results, making them impractical for everyday clinical use. Fortunately, wearable biosensors have emerged as promising non-invasive and real-time monitoring tools for musculoskeletal disorders [13,14,15]. These biosensors accurately measure physical and biological changes, including plantar pressure, and provide valuable information for the diagnosis and management of conditions. Furthermore, their portability makes them convenient for use in everyday clinical practice.

Here, a shoe-integrated sensor system (SISS) was proposed that utilizes sensors embedded in a shoe to detect plantar pressure during daily activities and provide objective and real-time data on ankle motion. Then, we prospectively utilized this system to measure the plantar pressure of CLAI patients and calculate the ratio of peak pressure between the unaffected and affected sides, and compared these results with those obtained through intraoperative arthroscopic examination of the distal tibiofibular joint, to conduct a diagnostic test. The purpose of this study was to determine the accuracy of the SISS as a promising new diagnostic tool for the detection of concomitant syndesmotic injury in individuals with CLAI.

## 2. Materials and Methods

### 2.1. Development of the Shoe-Integrated Sensor System

#### 2.1.1. Development of a Graphene-Based Force Sensing Resistive (FSR) Sensor

In the initial stage of our research, we developed a graphene-based flexible pressure sensor, which presents distinguished superiority in the pressure detection range and sensitivity [16]. The sensor is fabricated to be a sandwich structure that consists of two layers of Eco-flex that encapsulate a graphene oxide (GO) film possessing a laser-scribed graphene (LSG) sensing pattern as shown in Figure 1A. Ecoflex, a silicone rubber renowned for its remarkable elasticity and suppleness, was chosen as the substrate and encapsulation for the sandwich structure. Initially, a GO solution, featuring a volume ratio of 16.67% (volume ratio) tetrahydrofuran, was drop-cast onto the Ecoflex substrate surface to form a GO film after vaporization. The sample was then subjected to a commercial laser patterning device, which transformed the GO into LSG—essentially a multilayer graphene serving as the sensing component. To complete the encapsulation, connections were established using silver paste and copper wires, followed by a final layer of Ecoflex.

To enhance the accuracy of plantar pressure measurement, we conducted experiments with various parameter combinations for the FSR sensor. Eventually, we were able to create a set of sensors that had a solid square graphene sensing pattern with dimensions of 1 × 1 cm^2^ and a thickness of 2.5 mm (with a 1.25 mm Eco-flex layer on each side), achieving excellent sensitivity, linearity, and a wide range of detection. The unique characteristic of positive resistance shift against external pressure on the FSR sensing surface is demonstrated in Figure 1B. The positive resistance-pressure response makes it stand out from existing pressure sensors that are based on negative resistance variations. Owing to its great flexibility, high sensitivity, and broad detecting range, this sensor could be applied to monitor multiple physiological signals and physical motions [16].

#### 2.1.2. Method of Integration into the Shoe

A flexible printed circuit board (FPCB) was fabricated to form the insole substrate, establishing the sensor placement and connecting them to the data acquisition and transmission (DAT) modules. The pressure sensors were then integrated into the FPCB by welding and adhesion, ensuring secure placement. The designated sensor locations included the first phalanx (T1), first metatarsal (M1), third metatarsal (M3), fifth metatarsal (M5), medial cuneiform (medial midfoot, MM), tuberositas ossis metatarsalis V (lateral midfoot, ML), anterior calcaneus (anterior heel, HA), and posterior calcaneus (posterior heel, HP) (Figure 2A), where the sensor at medial cuneiform was set for users who have flat feet. To enhance reliability, a thin polyvinyl chloride film was placed atop the sensors and circuits to protect them without affecting their sensitivity to applied forces.

#### 2.1.3. System Hardware Information

The system hardware consists of two separate plantar pressure insoles and their corresponding DAT modules. These modules are capable of efficiently sampling and transmitting the plantar pressure data to the host computer, which is a smartphone. The design of the DAT module is compact, with a size of 35 mm by 35 mm. It is composed of an analog frontend (AFE), an analog-to-digital converter (ADC), a microprocessor, and a Bluetooth module, all of which are packed into a single CC2540R2f chip manufactured by Texas Instruments Inc. located in Texas, USA.

Each FSR sensor in the AFE is connected to a measuring resistor to perform the force-to-voltage conversion. The measuring resistor was chosen to optimize the desired force sensitivity range and to limit the current. The voltage divider is followed by an emitter follower and an amplifier implemented by a general operational amplifier chip (LMV324, Texas Instruments Inc., Dallas, TX, USA) and connected to an ADC with a sampling rate of 100 Hz. The average working current is around 20 mA, and a 500 mAh battery was chosen as the power supply, which is sufficient for a whole day’s usage.

With a thin and flexible insole and a compact DAT module, the overall hardware can be easily assembled into a regular shoe, as illustrated in Figure 2B. By adjusting the FPCB design according to the shoe sizes and sensor locations, our hardware can be well-suited for a variety of shoes, including sizes ranging from UK4 to UK10. As tested by users with different kinds of shoes, the overall system is comfortable, reliable, and suitable for daily plantar pressure sensing.

#### 2.1.4. System Workflow

Whenever a user initiates a walking test, the hardware part of SISS first detects the plantar pressure on both feet and transmits the data wirelessly to the analysis software in a smartphone in real-time. Then the program synchronizes the data from both feet and extracts and saves the valid plantar pressure data locally. After a walking test is finished, all the pressure changes are converted into the pressure-time curve and peak pressure is extracted and generated. At the same time, the software also contacts the cloud server and synchronizes the user data with its user database. Later, the users and their doctors can retrieve all their previous test data and analyze the results again.

### 2.2. Study Design

#### 2.2.1. Participants and Selection Criteria

All patients hospitalized for surgical intervention of ankle sprains between March 2022 and October 2022 were prospectively included. The cohort in the study consisted of individuals who had been diagnosed with CLAI and were suspected of having a syndesmotic injury. Participants were between the ages of 18 and 65 years old and did not have any significant comorbidities that might affect their gait or physical mobility. Patients who had prior surgical treatment of the lower extremity were excluded (Figure 3).

#### 2.2.2. Data Collection: Shoe-Integrated Sensor System

Participants were outfitted with the shoe-integrated sensor system, which consists of sensors embedded in a shoe. The sensors were calibrated to ensure accurate data collection. Participants were instructed to traverse a 15-m walkway at a pace of 1.3 m/s (3 steps per 2 s) to yield data on ankle motion. A trial was considered valid if it met the criteria of registering a heel strike pattern during the stance phase, with no discernible modifications in step length or step frequency that would affect the pressure plate. Each trial produced a time-peak force curve for eight regions of interest on the foot. The plantar pressure variables were calculated as the peak force normalized by body weight (kg/kg) for the specified region of interest [12]. The data was collected in real-time and securely stored in a database for later analysis.

#### 2.2.3. Arthroscopic Examination

Arthroscopic examination of the distal tibiofibular joint was performed during surgery utilizing a 30-degree 2.9 mm arthroscope (Smith & Nephew, Andover, MA, USA). Patients were positioned supine with the ankle joint in neutral. The standard anteromedial portal was used for observation, and the instrument was inserted into the joint through the anterolateral portal. After proper debridement of the synovial tissue, the ankle joint was distracted forward in neutral rotation, and the arthroscope was extended into the joint space to observe the syndesmosis. A custom-made probe with a diameter of 2–5 mm was used to detect whether the space of the syndesmosis was widened from the anterior to the posterior edge of the syndesmotic gap. The probe was stretched into the interval and the fibula side was pulled by the probe as the stress. The widths of the anterior, middle, and posterior sides were measured, and the width value in the middle interval was selected for analysis. The syndesmotic injury was diagnosed in cases of diastasis greater than 3 mm [10].

### 2.3. Statistical Analysis

The statistical analysis for this study was conducted using MedCalc Statistical Software version 20.1.0 (MedCalc Software bv, Ostend, Belgium), with a significance level of *p* < 0.05. Descriptive statistics were used to summarize the data, with continuous variables reported as mean ± standard deviation, and categorical variables reported as proportions.

To determine the optimal threshold for the predicted probability of a concomitant syndesmotic injury diagnosis, we used receiver operator characteristic (ROC) curve analysis. Logistic regression was utilized to build correlations between peak pressures in the seven plantar regions (independent variables) and the presence of syndesmotic injury (dependent variable). Predicted probabilities were obtained for each patient, based on their individual peak pressures, and used to plot receiver operator characteristic (ROC) curves. The ROC curves were generated using the predicted probabilities, with the highest Youden index as the cut-off value. The Youden index is a commonly used metric in ROC analysis that provides a balance between sensitivity and specificity [17] and identifies the optimal threshold for predicting the presence of syndesmotic injury.

To evaluate the diagnostic tests, we calculated sensitivity, specificity, positive and negative predictive values, and likelihood ratios (LRs) with 95% CIs. Diagnostic accuracy was assessed using Portney and Watkins’ method [18], whereby (true positive + true negative) was divided by the total number of cases. Likelihood ratios were considered to be clinically useful statistics [19,20] as well as the most reliable indicators of diagnostic accuracy if the incidence in the studied population is known [21]. LRs were interpreted according to Jaeschke’s guidelines [22], with values greater than 0.5 and less than 2 considered a very small or irrelevant change in likelihood, 0.2–0.5 or 2–5 considered a small change, 0.1–0.2 or 5–10 considered a moderate change, and <0.1 or >10 considered a large, often conclusive change in likelihood.

A likelihood nomogram was used to determine the probability that an individual similar to the participants of the present study would have concomitant syndesmotic injury using the established predicted probability threshold [23].

Overall, the statistical methods used in this analysis were designed to provide a comprehensive assessment of the accuracy of the diagnostic tests used and to determine the optimal threshold for predicting a concomitant syndesmotic injury diagnosis.

## 3. Results

### 3.1. Demographic Characteristic

Twenty-seven individuals (52% male) with chronic lateral ankle instability, averaging 29.4 ± 10.2 years of age, were successively recruited. At rest, the average level of discomfort was 2.9 ± 2.0 (ranging from 0 to 5) as opposed to 4.6 ± 1.5 (ranging from 2 to 8) during weight-bearing activities. The mean Beighton score was recorded at 2.2 (with a scale range of 0 to 7). The mean interval from the initial sprain to admission was 19 months (Table 1).

### 3.2. Plantar Pressure Measurements

Plantar pressure was measured by SISS for each region of interest (Table 2 and Appendix A), and there was a significant difference between the affected and the unaffected side in peak force in areas M1, M3, and M5 (*p* < 0.05). The between-limb peak pressure ratio was calculated and subjected to multivariate analysis via logistic regression, yielding results indicating a statistically significant association between the peak force ratio in the ML area and the presence of concomitant syndesmotic injury (*p* < 0.05).

### 3.3. ROC Curve and Cut-Off Point

Of the 27 participants, 15 were categorized as having concomitant syndesmotic injuries. Overall, the diagnostic accuracy for the test was moderate (78%, 95% CI 58–91%). The peak pressure ratio within the region of interest was analyzed through logistic regression to determine the predicted probability and establish a multivariate diagnostic model. The ROC curve was then plotted based on the predicted probabilities and the results of the arthroscopic diagnosis, and the optimal cut-off value was determined to be 0.51 (Figure 4). The area under the ROC curve was found to be 0.81 (*p* < 0.001, 95% CI 0.61–0.93), indicating that the SISS system had an 81% increased chance of accurately distinguishing between a CLAI case with syndesmosis diastasis and one without.

### 3.4. Sensitivity, Specificity, Positive and Negative Predictive Values, and Likelihood Ratios

The diagnostic efficacy of the SISS was presented in Table 3. By utilizing a cut-off value of a predicted probability exceeding 0.51, the sensitivity of the test was found to be 80% (95% CI 52–96%). This implies that the test accurately identified 80% of CLAI cases with a syndesmotic injury. Similarly, the specificity was found to be 75% (95% CI 43–95%), indicating that the test correctly identified 75% of CLAI cases without a syndesmotic injury. The positive predictive value was estimated at 80% (95% CI 59–92%), meaning that 80% of individuals who tested positive for the syndesmotic injury were confirmed to have it. On the other hand, the negative predictive value was 75% (95% CI 51–90%), meaning that 75% of individuals who tested negative for the condition were confirmed to be free of it. The positive likelihood ratio (LR+) reflects the degree to which the odds of having the condition increase when the test result is positive. In the case of this test, the LR+ was 3.20 (95% CI 1.16–8.80), indicating that the odds of having the syndesmotic injury are 3.20 times higher when the test result is positive. The negative likelihood ratio (LR–) reflects the degree to which the odds of having the condition decrease when the test result is negative. In this scenario, the LR– was 0.27 (95% CI 0.09–0.77), implying that the odds of having the syndesmotic injury are 0.27 times lower when the test result is negative.

Using the overall prevalence rate of syndesmotic injuries in this study (55.6%) as the estimated pretest probability, along with the calculated likelihood ratios, it was determined that the posttest probability of having a syndesmotic injury for a CLAI participant with a predicted probability greater than 0.51 was greater than 80% (95% CI 59–92%). Conversely, for a participant with a predicted probability less than 0.51, the posttest probability was 25% (95% CI 10–49%) (Figure 5).

## 4. Discussion

In summary, we demonstrated a shoe-integrated sensor system for providing real-time assessments of plantar pressure. By utilizing logistic regression to analyze the pressure data from various areas of the foot and through ROC curve analysis, our models achieved a noteworthy diagnostic capability for concomitant syndesmotic injuries in CLAI patients. With a sensitivity of 80% and a specificity of 75%, our models demonstrated a positive predictive value of 80% and a negative predictive value of 75%. The posttest probabilities indicated that a CLAI patient with a predicted probability exceeding 0.51 was more likely to have a syndesmotic injury (80%), while those with a predicted probability below 0.51 were less likely to have such an injury (25%).

The shoe-integrated sensor system consists of a thin and flexible smart insole with optimal sensor placements and a user-friendly app on the smartphone, coupled with a cloud-based data managing system, capable of accurately measuring plantar pressure in individuals during rest and motion. This system represents a quintessential utilization of biosensing technology in clinical medicine. Biosensing, which uses sensors and software to capture and analyze various physiological signals, holds the potential to provide real-time and continuous monitoring of patients. Biosensing technology has emerged as a promising tool for the diagnosis of orthopedic and sports medicine disorders. Luczak’s [24] development of a pressure-sensitive sock, employing compressible soft robotic sensors to measure plantar pressure, stand as a testament to this. Teramoto [25] utilized a capacitance-type sensor affixed to the skin near the anterior talofibular ligament to quantitatively evaluated anterior drawer laxity in patients with ankle instability. Wang’s [26] creation of a plantar pressure sensing array, complemented by a support vector machine supervised learning algorithm, served to enhance the diagnostic processes of lumbar degenerative disease. In this study, given the unique characteristics of plantar pressure in patients with ankle injuries, we devised a gait and plantar pressure detection mode and developed this SISS system using nanostructured sensors.

In our research, we crafted an FPCB to serve as the insole substrate, determining the positioning of sensors and linking them to the DAT modules. The pressure sensors were then affixed to the FPCB through welding and adhesion, guaranteeing their secure placement. In recent years, 3D electronic printing using an aerosol jet or inkjet printing has gained significant attention for its ability to fabricate high-performance and highly flexible electronic devices [27,28,29,30,31,32]. Techniques such as high-resolution aerosol jet printing [27] and inkjet printing of graphene-based flexible pressure sensors [29] have shown promising results in the development of stretchable electronics and wearable devices. However, such fabrication methods may not be robust enough for daily plantar pressure sampling applications, which require higher mechanical strength. While inkjet-printed soft resistive pressure sensor patches [28] and aerosol jet-printed multi-dimensional OECT force sensors [31] exhibit impressive sensitivity and flexibility, their durability and reliability might be compromised under the continuous mechanical stress that is experienced during daily plantar pressure sampling. Furthermore, the 3D printing of microfluidic sensors [30] has been investigated for soft robotics, but their application for insole substrates is still limited. To address these concerns and achieve a balance between flexibility and reliability, we chose to fabricate the insole circuits using a flexible printed circuit board (FPCB). The FPCB provides the necessary mechanical strength for daily use while maintaining a certain level of flexibility. Additionally, the use of FPCB allows for the potential of mass production, which can help reduce the overall system cost [27,29]. Another advantage of using FPCB in the insole substrate is the ease of sensor replacement. If a sensor is damaged, it can be easily replaced without replacing the entire insole. In contrast, 3D electronic printing often integrates the sensor, circuit, and substrate more closely, resulting in less flexibility for individual component replacement [27,28,29,31]. In summary, although 3D electronic printing using an aerosol jet or inkjet printing offers high performance and flexibility in fabricating electronic devices, its application for daily plantar pressure sampling requires a balance between flexibility and mechanical strength. By employing FPCB for the insole substrate, we can ensure greater reliability and ease of sensor replacement while still achieving acceptable flexibility and the potential for mass production.

The research disclosed that the shoe-integrated sensor system had a high rate of accuracy in differentiating CLAI patients with and without syndesmotic injuries, with a sensitivity of 80% and specificity of 75%. This meant that of all patients who had a concomitant syndesmotic injury, as determined by a gold standard method (arthroscopy), 80% of them were identified accurately by the SISS. Additionally, only 3 out of the 12 patients were wrongly classified as having an injury, as per the SISS readings. As per prior studies conducted by Sman, the conventional physical examination methods such as the dorsiflexion with external rotation stress test and squeeze test, had a sensitivity rate of up to 71% in the case of acute syndesmosis injury, but proved to be less effective in a chronic injury. De César’s study [33], on the other hand, reported that the sensitivity of physical examination was only between 20–30%. However, the specificity of physical examination was recorded to be 93.5%, which is higher compared to the 75% specificity of the SISS. Despite the high sensitivity of the SISS, which makes it a suitable tool for initial screenings, it is crucial to note that for patients with suspected syndesmotic injuries, a combination of physical examination tests is still necessary to arrive at a final diagnosis and avoid any misdiagnosis due to the system’s relatively low specificity rate. The combination of the SISS system and physical examination should be studied in the future.

The positive predictive value suggested that 80% of the participants diagnosed with a concomitant syndesmotic injury through plantar pressure analysis were verified through arthroscopy. The negative predictive value of 75% revealed that three-quarters of those with uninjured syndesmotic joints through plantar pressure measurements were normal in terms of distal tibiofibular width. According to Großterlinden [34], the positive predictive value of palpation and special physical examination was only 16.7% to 44.4%, while the negative predictive value was between 59.6% and 86.7%. The results of the SISS demonstrate a superior diagnostic value compared to physical examination. Nevertheless, the incidence of syndesmotic injuries in ankle sprains was 10–20% [35,36], while it was 56% in the included cases of CLAI. The trauma mechanism of dorsiflexion compounded with rotational force is likely to inflict harm upon both the lateral ankle ligament and the distal tibiofibular joint [4,5], leading to a relatively elevated prevalence of injury in the study population. While an increase in incidence would result in an increase in predictive values, it is important to emphasize that the predictive value of this system would decrease in actual clinical practice.

The positive likelihood ratio was computed at 3.20, and post-test probability analysis reveals that if a CLAI patient’s predicted probability exceeds 0.51 based on the between-limb plantar pressure ratio derived from the SISS, there is an 80% probability of concomitant syndesmotic injury. The negative likelihood ratio of 0.27 also implies that if the predicted probability is less than 0.51, there is only a 25% chance of actual syndesmotic injury. These findings suggested that the newly developed diagnostic model can effectively identify and exclude patients with syndesmotic injuries. Given that the presence of syndesmotic injury was associated with an approximately six-fold increase in the time to return to activities and a significant increase in the risk of substantial arthritis compared to isolated ankle sprains [37], the utilization of the SISS system could result in a more accurate diagnosis of syndesmotic injury, and subsequently, the provision of more appropriate treatment recommendations including surgery to correct multiple instabilities.

The present study is subject to several limitations. Firstly, the cohort of patients recruited for this study constituted a relatively small sample size of largely homogenous individuals and was inadequate in comparison to the diversity of the variables analyzed. Secondly, the diagnostic models were developed using solely the dataset as the training set and no external validation was conducted on the model with a separate population. Additionally, gait patterns in everyday life encompass a multitude of movements such as running, cutting, or landing, and it cannot be presupposed that the results of this study are generalizable to other forms of physical activity beyond walking. To bolster the validity of the study, future research should aim to incorporate larger and more diverse patient samples, examine a wider array of exercise modalities, and validate the diagnostic efficacy of the models by applying them to independent populations.

## 5. Conclusions

The shoe-integrated sensor system devised in the current study exhibits remarkable precision in identifying concomitant syndesmotic injuries in patients with chronic lateral ankle instability. By evaluating the between-limb plantar pressure ratio while walking with logistic regression predicted value > 0.51, the system exhibits a high level of sensitivity (80%) and specificity (75%) in determining the presence of syndesmotic injury, with posttest probabilities of 80% and 25% for positive and negative results, respectively. The non-invasive and user-friendly nature of the SISS system represents a promising innovation in the diagnosis of syndesmotic injury in CLAI cases. These results lay the groundwork for incorporating wearable sensors into clinical practice, as they furnish invaluable data on gait patterns and physical activity, leading to accurate diagnoses and optimized treatment plans.

## Figures and Tables

**Figure 1 nanomaterials-13-01539-f001:**
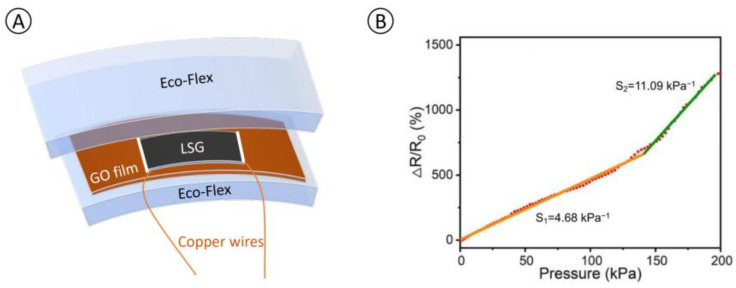
(**A**): Schematic layout of a single sensor with an LSG pattern, two wires, and a vertically sandwiched Eco-flex capsulation; (**B**): Relative resistance variations of the FSR sensor with a 1 × 1 cm^2^ solid square sensing pattern under external pressure along the *z* direction.

**Figure 2 nanomaterials-13-01539-f002:**
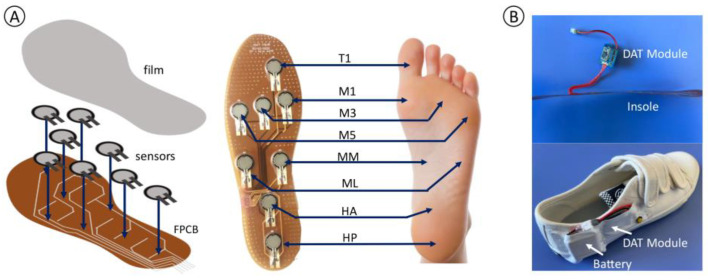
(**A**): The structure of the pressure detection insole and the definition of the sensor locations; (**B**): The display of the DAT module, the insole, and the overall hardware in a shoe.

**Figure 3 nanomaterials-13-01539-f003:**
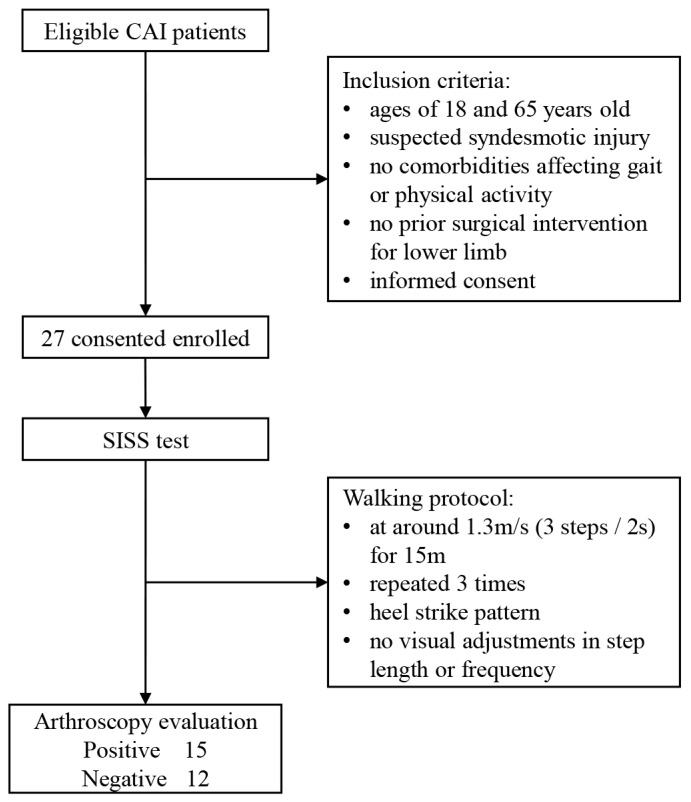
A flow diagram of the study.

**Figure 4 nanomaterials-13-01539-f004:**
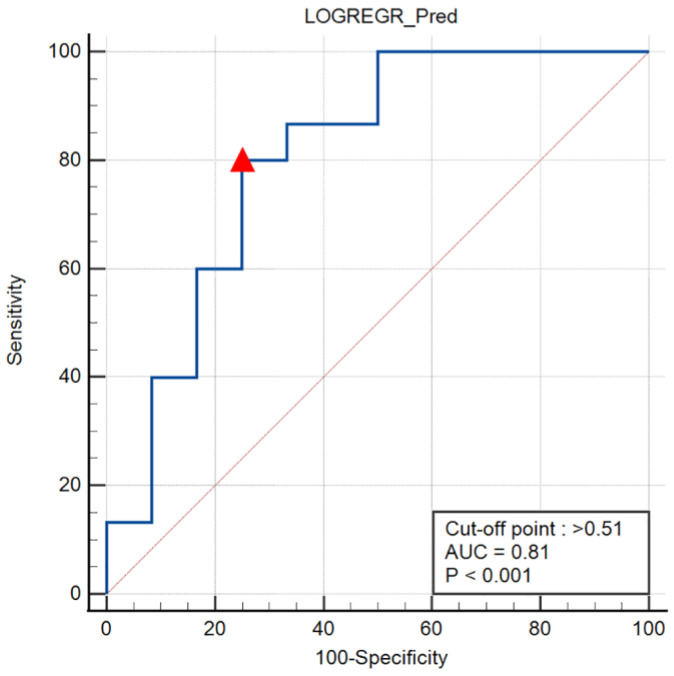
Receiver operator characteristic curve for SISS diagnostic accuracy. The cut-point of >0.51 is used to denote likely syndesmotic injury.

**Figure 5 nanomaterials-13-01539-f005:**
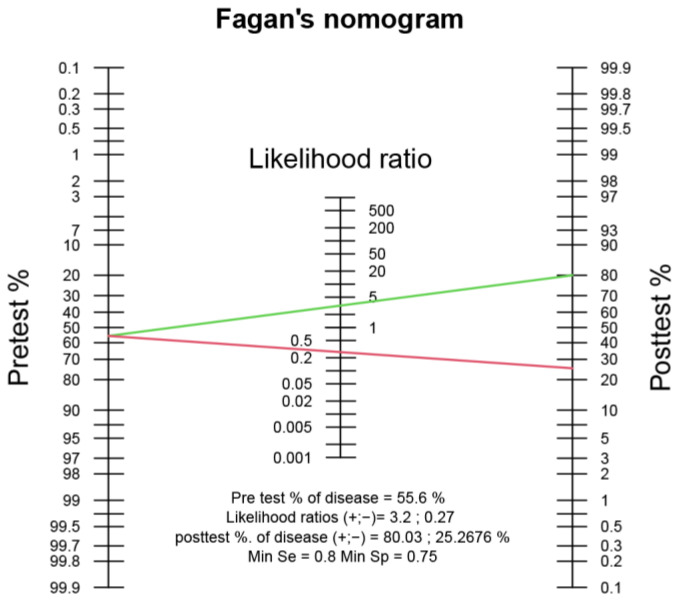
A likelihood ratio nomogram. The pretest probability of concomitant syndesmotic injury in CLAI patients was estimated at 55.6%. The LR+ of 3.20 for the predicted probability of the SISS system was indicated, alongside the posttest probability of 80% (green line). The LR– of 0.27 was indicated, along with the posttest probability of 25% (red line).

**Table 1 nanomaterials-13-01539-t001:** Patient Characteristics.

	*n* = 27
Age (yr)	29.4 ± 10.2
Gender	
Male	14 (52%)
Female	13 (48%)
Body mass index (kg/m^2^)	24.9 ± 4.2
Height (cm)	170.1 ± 9.9
Weight (kg)	73.1 ± 15.9
Beighton score	2.2 (0–7)
Pain	
At rest (VAS 0–10)	2.9 ± 2.0
Under weight-bearing (VAS 0–10)	4.6 ± 1.5
Post-injury duration (mo)	19 (3–72)

**Table 2 nanomaterials-13-01539-t002:** Plantar pressure measurements of SISS.

		Peak Force			Ratio	
	Affected	Unaffected	*p*	Syndesmotic Injury Group	Control Group	*p*
T1	0.12 ± 0.06	0.13 ± 0.05	0.20	0.97 ± 0.35	0.97 ± 0.37	0.66
M1	0.16 ± 0.05	0.20 ± 0.05	0.01 *	0.81 ± 0.22	0.88 ± 0.43	0.24
M3	0.23 ± 0.05	0.24 ± 0.04	0.01 *	0.92 ± 0.14	0.97 ± 0.08	0.13
M5	0.15 ± 0.03	0.16 ± 0.03	0.04 *	0.93 ± 0.19	0.97 ± 0.12	0.32
ML	0.06 ± 0.03	0.07 ± 0.03	0.11	0.85 ± 0.27	1.10 ± 0.40	0.04 *
HA	0.08 ± 0.04	0.08 ± 0.04	0.77	1.08 ± 0.32	1.05 ± 0.37	0.52
HP	0.20 ± 0.04	0.20 ± 0.05	0.10	1.01 ± 0.11	0.98 ± 0.15	0.45

* *p* < 0.05.

**Table 3 nanomaterials-13-01539-t003:** Diagnostic test accuracy of SISS.

	SISS Test Accuracy (95% CI)
Diagnostic accuracy	78% (95% CI 58–91%)
Sensitivity	80% (95% CI 52–96%)
Specificity	75% (95% CI 43–95%)
Positive predictive value	80% (95% CI 59–92%)
Negative predictive value	75% (95% CI 51–90%)
LR+	3.20 (95% CI 1.16–8.80)
LR−	0.27 (95% CI 0.09–0.77)

## Data Availability

Not applicable.

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
