# Peer review of "Shoe-Integrated Sensor System for Diagnosis of the Concomitant Syndesmotic Injury in Chronic Lateral Ankle Instability: A Prospective Double-Blind Diagnostic Test"

_nanomaterials, 2023, doi:10.3390/nano13091539_

Round 1

Reviewer 1 Report

The authors analysed the CLAI by means of a shoe integrated sensor system. In my opinion, more technical information about the system should be included such us:

Details or electronic parts: Sensor signal conditioning, how the signal is transmitted etc..

Battery lifetime

Shoe comfort

Author Response

Response to Reviewer 1

[General comment] The authors analysed the CLAI by means of a shoe integrated sensor system. In my opinion, more technical information about the system should be included such as: Details or electronic parts: Sensor signal conditioning, how the signal is transmitted etc..; Battery lifetime; Shoe comfort.

Response: Thank you very much for the comment. We agree that we should report more technical details about the SISS, and we add relevant information, including the composition of the DAT, chip model, sensor signal conditioning, signal transmitting, sampling frequency, operating current, battery lifetime, and shoe comfort. We added sections as follows:

“2.1.3 System hardware information

The system hardware consists of two separate plantar pressure insoles and their corresponding DAT modules. These modules are capable of efficiently sampling and transmitting the plantar pressure data to the host computer, which is a smartphone. The design of the DAT module is compact, with a size of 35 mm by 35 mm. It is composed of an analog frontend (AFE), an analog-to-digital converter (ADC), a microprocessor, and a Bluetooth module, all of which are packed into a single CC2540R2f chip manufactured by Texas Instruments Inc. located in Texas, USA.

Each FSR sensor in the AFE is connected to a measuring resistor to perform the force-to-voltage conversion. The measuring resistor was chosen to optimize the desired force sensitivity range and to limit the current. The voltage divider is followed by an emitter follower and an amplifier implemented by a general operational amplifier chip (LMV324, Texas Instruments Inc.) and connected to an ADC with a sampling rate of 100 Hz. The average working current is around 20 mA, and a 500 mAh battery was chosen as the power supply, which is sufficient for a whole day's usage.

With a thin and flexible insole and a compact DAT module, the overall hardware can be easily assembled into a regular shoe, as illustrated in Figure 2B. By adjusting the FPCB design according to the shoe sizes and sensor locations, our hardware can be well-suited for a variety of shoes, including sizes ranging from UK4 to UK10. As tested by users with different kinds of shoes, the overall system is comfortable, reliable, and suitable for daily plantar pressure sensing.

2.1.4 System workflow

Whenever a user initiates a walking test, the hardware part of SISS first detects the plantar pressure on both feet and transmits the data wirelessly to the analysis software in smartphone in real-time. Then the program synchronizes the data from both feet and extracts and saves the valid plantar pressure data locally. After a walking test is finished, all the pressure changes are converted into the pressure-time curve and peak pressure is extracted and generated. At the same time, the software also contacts the cloud server and synchronizes the user data with its user database. Later, the users and their doctors can retrieve all their previous test data and analyze the results again.” [Line 125-154]

Reviewer 2 Report

The reviewed manuscript describes possible investigation of chronic lateral ankle instability by means of shoe-integrated sensor system. The topic is interested but there are some weak points in the papers to be improved before possible publication in Nanomaterials:

1.There is no information about the parameters of the sensors used in the measurements.

2. Too cursory description of statistical analysis.

3. A small statistical sample of the examined patients.

Wyniki tłumaczenia

Tłumaczenie

4, The authors did not explain the parameters obtained by statistical reasoning.

5. Some formal errors need to be corrected, such as: 

- e.g. please write Nyska [1] (with a space) and not Nyska[1],

- please begin subsection titles with a capital letter (eg. Method ofintegration ...),

- please write eg. 3 mm (with space) - not 3 mm.

Based on above remarks I think that revision of this manuscript is necessary.

Author Response

Point-by-point response to the comments from the reviewers

Response to Reviewer 2

[General comment] The reviewed manuscript describes possible investigation of chronic lateral ankle instability by means of shoe-integrated sensor system. The topic is interested but there are some weak points in the papers to be improved before possible publication in Nanomaterials.

Response: Thank you for your review of our manuscript. We appreciate your interest in our work, and we agree that there are some areas that require improvement before possible publication. We plan to strengthen our paper by providing more experimental details and describing statistical analyses more thoroughly. We hope that the manuscript has been improved after this revision.

[Comment 1] There is no information about the parameters of the sensors used in the measurements.

Response: Thank you very much for the comment. The relevant parameters of the sensors used were supplemented. We added sentences and figure as follows:

“To enhance the accuracy of plantar pressure measurement, we conducted experiments with various parameter combinations for the FSR sensor. Eventually, we were able to create a set of sensors that had a solid square graphene sensing pattern with dimensions of 1 × 1 cm2 and a thickness of 2.5 mm (with a 1.25 mm Eco-flex layer on each side), achieving excellent sensitivity, linearity, and a wide range of detection. The unique characteristic of positive resistance shift against external pressure on the FSR sensing surface is demonstrated in Figure 1B.” [Line 95-101]

Figure 1B: Relative resistance variations of the FSR sensor with a 1 × 1 cm2 solid square sensing pattern under external pressure along the z direction. [Line 105-109]

[Comment 2] Too cursory description of statistical analysis.

Response: Thanks for the comment. We expanded the initial description of the statistical analysis, provided more details on the specific methods used for analyzing the data, and added more information on how the results were interpreted. We revised the section as follows:

“The statistical analysis for this study was conducted using MedCalc Statistical Software version 20.1.0 (MedCalc Software bv, Ostend, Belgium), with a significance level of P <0.05. Descriptive statistics were used to summarize the data, with continuous var-iables reported as mean ± standard deviation, and categorical variables reported as proportions.

To determine the optimal threshold for the predicted probability of a concomitant syndesmotic injury diagnosis, we used receiver operator characteristic (ROC) curve analysis. Logistic regression was utilized to build correlations between peak pressures in the seven plantar regions (independent variables) and the presence of syndesmotic injury (dependent variable). Predicted probabilities were obtained for each patient, based on their individual peak pressures, and used to plot receiver operator characteristic (ROC) curves. The ROC curves were generated using the predicted probabilities, with the highest Youden index as the cut-off value. The Youden index is a commonly used metric in ROC analysis that provides a balance between sensitivity and specificity [17], and identifies the optimal threshold for predicting the presence of syndesmotic injury.

To evaluate the diagnostic tests, we calculated sensitivity, specificity, positive and negative predictive values, and likelihood ratios (LRs) with 95% CIs. Diagnostic accuracy was assessed using Portney and Watkins’ method [18], whereby (true positive + true negative) was divided by the total number of cases. Likelihood ratios were considered to be clinically useful statistics [19,20] as well as the most reliable indicators of diagnostic accuracy if the incidence in the studied population is known [21]. LRs were interpreted according to Jaeschke's guidelines [22], with values greater than 0.5 and less than 2 considered a very small or irrelevant change in likelihood, 0.2-0.5 or 2-5 considered a small change, 0.1-0.2 or 5-10 considered a moderate change, and < 0.1 or > 10 considered a large, often conclusive change in likelihood.

A likelihood nomogram was used to determine the probability that an individual similar to the participants of the present study would have concomitant syndesmotic injury using the established predicted probability threshold [23].

Overall, the statistical methods used in this analysis were designed to provide a comprehensive assessment of the accuracy of the diagnostic tests used, and to determine the optimal threshold for predicting a concomitant syndesmotic injury diagnosis.” [Line 190-219]

[Comment 3] A small statistical sample of the examined patients.

Response: Thank you for your comment. It is true that the sample size in our study is relatively small, which is one of the limitations of this study. However, before conducting the logistic regression analysis, we performed a univariate analysis of the seven independent variables, and eventually, only one variable (ML, as shown in Table 2) was found to be statistically significant. Additionally, the incidence of distal tibiofibular injury in chronic ankle instability (CAI) patients is about 50%. Based on the recommended events per variable (EPV) method estimation, assuming EPV = 10, the required sample size should be 10 * 1 / 50% = 20. In our study, we included 27 patients, which exceeded the minimum sample size recommended by the EPV method. We included seven variables in our analysis to adjust for confounding brought about by other plantar pressure variables. While we understand the limitation of our sample size, we believe that our study still provides valuable insights into the relationship between plantar pressures and syndesmotic injury. In the future, we plan to include more samples in our study to develop more accurate models containing multiple covariates.

[Comment 4] The authors did not explain the parameters obtained by statistical reasoning.

Response: Thank you for your comment. We supplemented the description and interpretation of the statistical reasoning related to the accuracy of the test. We revised and added sentences as follows:

“By utilizing a cut-off value of a predicted probability exceeding 0.51, the sensitivity of the test was found to be 80% (95% CI 52-96%). This implies that the test accurately identified 80% of CLAI cases with a syndesmotic injury. Similarly, the specificity was found to be 75% (95% CI 43-95%), indicating that the test correctly identified 75% of CLAI cases without a syndesmotic injury. The positive predictive value was estimated at 80% (95% CI 59-92%), meaning that 80% of individuals who tested positive for the syndesmotic injury were confirmed to have it. On the other hand, the negative predictive value was 75% (95% CI 51-90%), meaning that 75% of individuals who tested negative for the condition were confirmed to be free of it. The positive likelihood ratio (LR+) reflects the degree to which the odds of having the condition increase when the test result is positive. In the case of this test, the LR+ was 3.20 (95% CI 1.16-8.80), indicating that the odds of having the syn-desmotic injury are 3.20 times higher when the test result is positive. The negative likelihood ratio (LR-) reflects the degree to which the odds of having the condition de-crease when the test result is negative. In this scenario, the LR- was 0.27 (95% CI 0.09-0.77), implying that the odds of having the syndesmotic injury are 0.27 times lower when the test result is negative.” [Line 254-269]

[Comment 5] Some formal errors need to be corrected, such as:

- e.g. please write Nyska [1] (with a space) and not Nyska[1],

- please begin subsection titles with a capital letter (eg. Method of integration ...),

- please write eg. 3 mm (with space) - not 3mm.

Response: Thank you very much for the reminder. We have made revisions accordingly.

Reviewer 3 Report

1. The authors mentioned that the sensor used in this work is based on one of the author's previous work. However, the dimension of the sensor such as the thickness of ECOflex material is not discussed in this manuscript and the other published work. Suggest disclosing the thickness of the sensor and the thickness of the ECOflex can potentially affecting the sensitivity of the sensor.

2. It is not clear to the reader how the SSIS works. Suggest adding a section to elaborate more on the test.

3. Suggest providing an example of the SSIS outputs of a negative and positive case of CLAI.

4. Since this journal is about nanomaterial, suggest adding some more information related to nanomaterial to justify the suitability of this manuscript to the journal.

5. Can the authors also discuss in the manuscript why the authors chose to fabricate the sensor and circuit that way? Why not other method such as 3d electronic printing using aerosol jet or inkjet printing? Suggest discussing, comparing, and citing all the papers below to enrich the discussion.

a.  G. L. Goh, S. Agarwala, and W. Y. Yeong: 'High Resolution Aerosol Jet Printing of Conductive Ink for Stretchable Electronics', Proceedings of the 3rd International Conference on Progress in Additive Manufacturing (PRO-AM), Nanyang Technological University, Singapore, 2018, 109-114.

b. Lo, L. W., Shi, H., Wan, H., Xu, Z., Tan, X., & Wang, C. (2020). Inkjet‐printed soft resistive pressure sensor patch for wearable electronics applications. Advanced Materials Technologies5(1), 1900717.

c. Sun, J., Sun, Y., Jia, H., Bi, H., Chen, L., Que, M., ... & Sun, L. (2022). A novel pre-deposition assisted strategy for inkjet printing graphene-based flexible pressure sensor with enhanced performance. Carbon196, 85-91.

d. Goh, G. L., Agarwala, S., & Yeong, W. Y. (2016). 3D printing of microfluidic sensor for soft robots: a preliminary study in design and fabrication. Proceedings of the 2nd International Conference on Progress in Additive Manufacturing (Pro‑AM 2016), 177‑181.

e. Zhou, X., Zhang, L., Wang, Y., Zhao, S., Zhou, Y., Guo, Y., ... & Chen, H. (2023). Aerosol Jet Printing of Multi‐Dimensional OECT Force Sensor with High Sensitivity and Large Measuring Range. Advanced Materials Technologies, 2201272.

f. Blumenthal, T., Fratello, V., Nino, G., & Ritala, K. (2013, April). Conformal printing of sensors on 3D and flexible surfaces using aerosol jet deposition. In Nanosensors, Biosensors, and Info-Tech Sensors and Systems 2013 (Vol. 8691, pp. 118-126). SPIE.

Author Response

Point-by-point response to the comments from the reviewers

Response to Reviewer 3

[Comment 1] The authors mentioned that the sensor used in this work is based on one of the author's previous work. However, the dimension of the sensor such as the thickness of ECOflex material is not discussed in this manuscript and the other published work. Suggest disclosing the thickness of the sensor and the thickness of the ECOflex can potentially affecting the sensitivity of the sensor.

Response: Thank you very much for your nice reminder. We supplemented the thickness of the sensor and the thickness of the ECOflex. We added sentences as follows:

“Eventually, we were able to create a set of sensors that had a solid square graphene sensing pattern with dimensions of 1 × 1 cm2 and a thickness of 2.5 mm (with a 1.25 mm Eco-flex layer on each side), achieving excellent sensitivity, linearity, and a wide range of detection.” [Line 96-99]

[Comment 2] It is not clear to the reader how the SSIS works. Suggest adding a section to elaborate more on the test.

Response: Thank you very much for the comment. We added a a section to elaborate how the SSIS works as follows:

“2.1.4 System workflow

Whenever a user initiates a walking test, the hardware part of SISS first detects the plantar pressure on both feet and transmits the data wirelessly to the analysis software in smartphone in real-time. Then the program synchronizes the data from both feet and extracts and saves the valid plantar pressure data locally. After a walking test is finished, all the pressure changes are converted into the pressure-time curve and peak pressure is extracted and generated. At the same time, the software also contacts the cloud server and synchronizes the user data with its user database. Later, the users and their doctors can retrieve all their previous test data and analyze the results again.” [Line 146-154]

[Comment 3] Suggest providing an example of the SSIS outputs of a negative and positive case of CLAI.

Response: Thank you for your comment. However, the SISS is a measurement tool for plantar pressure, and it currently does not provide positive or negative judgments. Instead, it outputs a table with peak pressure values for each region (as shown in Table S1 below), which we use to calculate the between-limb peak pressure ratio. We established the relationship between syndesmotic injury and plantar pressure by logistic regression and finally obtained the predicted probability of the rightmost column in the table. In this study, we identified the best threshold for predictive probability by ROC curve analysis, which demonstrated the feasibility of using the SISS in this regard. We plan to add a positive/negative results function to the SISS software interface once we have included more variables to improve the model's accuracy.

Table S1. Peak pressure ratio measured by SISS system and calculated predicted probabilities in 27 CLAI patients.

Disease: 0 = normal syndesmosis; 1 = syndesmotic injury; LOGREGR_Pred: predicted probability calculated by logistic regression; NA: not applicable.

[Comment 4] Since this journal is about nanomaterial, suggest adding some more information related to nanomaterial to justify the suitability of this manuscript to the journal.

Response: Thanks for the comment. We supplement the process of sensor fabrication and encapsulation as follows:

“Ecoflex, a silicone rubber renowned for its remarkable elasticity and suppleness, was chosen as the substrate and encapsulation for the sandwich structure. Initially, a GO solution, featuring a volume ratio of 16.67% (volume ratio) tetrahydrofuran, was drop-cast onto the Ecoflex substrate surface to form a GO film after vaporization. The sample was then subjected to a commercial laser patterning device, which transformed the GO into LSG - essentially a multilayer graphene serving as the sensing component. To complete the encapsulation, connections were established using silver paste and copper wires, followed by a final layer of Ecoflex.” [Line 87-94]

[Comment 5] Can the authors also discuss in the manuscript why the authors chose to fabricate the sensor and circuit that way? Why not other method such as 3d electronic printing using aerosol jet or inkjet printing? Suggest discussing, comparing, and citing all the papers below to enrich the discussion.

  1. G. L. Goh, S. Agarwala, and W. Y. Yeong: 'High Resolution Aerosol Jet Printing of Conductive Ink for Stretchable Electronics', Proceedings of the 3rd International Conference on Progress in Additive Manufacturing (PRO-AM), Nanyang Technological University, Singapore, 2018, 109-114.
  2. Lo, L. W., Shi, H., Wan, H., Xu, Z., Tan, X., & Wang, C. (2020). Inkjet‐printed soft resistive pressure sensor patch for wearable electronics applications. Advanced Materials Technologies, 5(1), 1900717.
  3. Sun, J., Sun, Y., Jia, H., Bi, H., Chen, L., Que, M., ... & Sun, L. (2022). A novel pre-deposition assisted strategy for inkjet printing graphene-based flexible pressure sensor with enhanced performance. Carbon, 196, 85-91.
  4. Goh, G. L., Agarwala, S., & Yeong, W. Y. (2016). 3D printing of microfluidic sensor for soft robots: a preliminary study in design and fabrication. Proceedings of the 2nd International Conference on Progress in Additive Manufacturing (Pro‑AM 2016), 177‑181.
  5. Zhou, X., Zhang, L., Wang, Y., Zhao, S., Zhou, Y., Guo, Y., ... & Chen, H. (2023). Aerosol Jet Printing of Multi‐Dimensional OECT Force Sensor with High Sensitivity and Large Measuring Range. Advanced Materials Technologies, 2201272.
  6. Blumenthal, T., Fratello, V., Nino, G., & Ritala, K. (2013, April). Conformal printing of sensors on 3D and flexible surfaces using aerosol jet deposition. In Nanosensors, Biosensors, and Info-Tech Sensors and Systems 2013 (Vol. 8691, pp. 118-126). SPIE.

Response: Thank you. It’s a good question. 3d electronic printing using aerosol jet or inkjet printing is known for its ability to fabricate high performance and high flexible electronic devices. However, such a way of fabricating circuit may not be robust enough to be used in the daily plantar pressure sampling. In this circumstance, we need a less flexible but a more reliable way of fabricating the insole circuits, which requires a higher mechanical strength. Also, to reduce the cost of the overall system and increase the potential probility to do massive production, the flexible printed circuit board (FPCB) was eventually used in the insole substrate. In addition, if a sensor is damaged, it can be easily replaced without replacing the whole insole. 3d electronic printing often integrates sensor, circuit, and substrate more closely but with less flexibility and therefore may not be suitable in this circumstance. As suggested, certain discussions and citations are added in the paper as follows:

“In our research, we crafted a FPCB to serve as the insole substrate, determining the positioning of sensors and linking them to the DAT modules. The pressure sensors were then affixed to the FPCB through welding and adhesion, guaranteeing their secure placement. In recent years, 3D electronic printing using aerosol jet or inkjet printing has gained significant attention for its ability to fabricate high-performance and highly flexible electronic devices [27–32]. Techniques such as high-resolution aerosol jet printing [27] and inkjet printing of graphene-based flexible pressure sensors [29] have shown promising results in the development of stretchable electronics and wearable devices. However, such fabrication methods may not be robust enough for daily plantar pressure sampling applications, which require higher mechanical strength. While inkjet-printed soft resistive pressure sensor patches [28] and aerosol jet printed multi-dimensional OECT force sensors [31] exhibit impressive sensitivity and flexibility, their durability and re-liability might be compromised under the continuous mechanical stress that is experi-enced during daily plantar pressure sampling. Furthermore, 3D printing of microfluidic sensors [30] has been investigated for soft robotics, but their application for insole sub-strates is still limited. To address these concerns and achieve a balance between flexibility and reliability, we chose to fabricate the insole circuits using a flexible printed circuit board (FPCB). The FPCB provides the necessary mechanical strength for daily use while maintaining a certain level of flexibility. Additionally, the use of FPCB allows for the potential of mass production, which can help reduce the overall system cost [27,29]. Another advantage of using FPCB in the insole substrate is the ease of sensor replacement. If a sensor is damaged, it can be easily replaced without replacing the entire insole. In contrast, 3D electronic printing often integrates the sensor, circuit, and substrate more closely, resulting in less flexibility for individual component replacement [27–29,31]. In summary, although 3D electronic printing using aerosol jet or inkjet printing offers high performance and flexibility in fabricating electronic devices, its application for daily plantar pressure sampling requires a balance between flexibility and mechanical strength. By employing FPCB for the insole substrate, we can ensure greater reliability and ease of sensor replacement while still achieving acceptable flexibility and the potential for mass production.” [Line 308-336]

Round 2

Reviewer 1 Report

My suggestion has been addressed. I recommend the paper to be published.

Reviewer 2 Report

All doubts were explained and weak points of the manuscript were corrected.

I propose to publish this paper essentially as is.

Reviewer 3 Report

The authors have addressed the queries adequately, the manuscript can therefore be recommended for acceptance for publication.